# Genomic and Transcriptomic Survey Provides Insights into Molecular Basis of Pathogenicity of the Sunflower Pathogen *Phoma macdonaldii*

**DOI:** 10.3390/jof9050520

**Published:** 2023-04-27

**Authors:** Xuejing Chen, Xiaoran Hao, Oren Akhberdi, Xudong Zhu

**Affiliations:** 1College of Biological and Geography Sciences, Yili Normal University, Yining 835000, China; xuejingchen97@163.com; 2National Experimental Teaching Demonstrating Center, College of Life Sciences, Beijing Normal University, Beijing 100875, China; 2015xrhao@bnu.edu.cn; 3Key Laboratory of Microbial Resources Protection, Development and Utilization, Yili Normal University, Yining 835000, China; 4Beijing Key Laboratory of Genetic Engineering Drug and Biotechnology, College of Life Sciences, Beijing Normal University, Beijing 100875, China

**Keywords:** *Phoma macdonaldii*, sunflower, black stem, virulence genes, transcriptome, whole-genome sequencing

## Abstract

*Phoma macdonaldii* (teleomorph *Leptosphaeria lindquistii*) is the causal agent of sunflower (*Helianthus annuus* L.) black stem. In order to investigate the molecular basis for the pathogenicity of *P. ormacdonaldii*, genomic and transcriptomic analyses were performed. The genome size was 38.24 Mb and assembled into 27 contigs with 11,094 putative predicted genes. These include 1133 genes for CAZymes specific for plant polysaccharide degradation, 2356 for the interaction between the pathogen and host, 2167 for virulence factors, and 37 secondary metabolites gene clusters. RNA-seq analysis was conducted at the early and late stages of the fungal spot formation in infected sunflower tissues. A total of 2506, 3035, and 2660 differentially expressed genes (DEGs) between CT and each treatment group (LEAF-2d, LEAF-6d, and STEM) were retrieved, respectively. The most significant pathways of DEGs from these diseased sunflower tissues were the metabolic pathways and biosynthesis of secondary metabolites. Overall, 371 up-regulated DEGs were shared among LEAF-2d, LEAF-6d, and STEM, including 82 mapped to DFVF, 63 mapped to PHI-base, 69 annotated as CAZymes, 33 annotated as transporters, 91 annotated as secretory proteins, and a carbon skeleton biosynthetic gene. The most important DEGs were further confirmed by RT-qPCR. This is the first report on the genome-scale assembly and annotation for *P. macdonaldii*. Our data provide a framework for further revealing the underlying mechanism of the pathogenesis of *P. macdonaldii*, and also suggest the potential targets for the diseases caused by this fungal pathogen.

## 1. Introduction

Sunflower (*Helianthus annuus* L.) is the fourth most important oilseed crop ranked after soybeans, rapeseed, and safflower, and is cultivated in more than 70 countries and regions worldwide, with an annual yield of more than 50 million tons [1]. Globally, the sunflower crop is faced with several diseases caused by fungi and oomycetes in all the regions where it is cultivated [2]. With the increasing production and global trend of sunflowers, sunflower diseases have become a significant concern and indeed have already caused serious economic losses [3]. Black stem caused by the necrotrophic fungus *Phoma macdonaldii* (teleomorph *Leptosphaeria lindquistii*) is one of the most devastating sunflower diseases. This disease can be seed born, spreads rapidly, and occurs at all sunflower growth stages [4]. The characteristic symptoms of black stem are black oval to long lesions on the petiole and stems [5], which can cause early maturity and death in sunflowers [6]. The black stem of sunflower was first detected in Canada in 1964 but now occurs in Asia, Africa, North and South America, and Europe [7]. China is the fifth largest sunflower-producing country in the world [8]. In China, sunflowers are widely grown in Xinjiang, Gansu, Inner Mongolia, Ningxia, Shanxi, Heilongjiang, Jilin, and Hebei provinces [9]. The occurrence of sunflower black stem in China was first reported in 2008 in the Ili Valley of the Xinjiang Uygur Autonomous Region and 49% of the sunflower fields in the area were infected with *P. macdonaldii* [10]. In 2010, black stem was added to the quarantine sunflower disease list in China.

Controlling the black stem caused by *P. macdonaldii* is challenging. Over the past decades, extensive efforts have been made to identify the genes involved in the partial resistance of sunflower to *P. macdonaldii* via plant–microbe interactions, and the resistant sunflower varieties that can be used in various plant breeding programs [11,12,13,14]. On the other side, numerous phytotoxins with various structures have been isolated and identified from the *Phoma* species [15]. To date, the morphological and physiological characteristics of *P. macdonaldii* have been described, and the varying aggressiveness of *P. macdonaldii* isolates has been evaluated, which provided the fundamental details regarding the pathogenicity of this fungus [16,17]. However, only a few *Phoma* spp. sequences (mostly derived from ITS regions) are deposited in the NCBI database, and the molecular mechanisms involved in *Phoma* (especially for *P. macdonaldii*) pathogenicity/virulence and host–pathogen interactions are still unclear. Recently, a global proteomic analysis of *P. macdonaldii* has been conducted [18]. A total of 1498 proteins were identified, roughly classified into the three main GO categories, or annotated into the five major KEGG databases. The homologues gene expression of CK1, Ras-1, Mth, 14-3-3, NADPH-cytochrome P450 reductase (P450), peptidyl-prolyl cis-trans isomerases (PPIases), and Nac that previously reported to be related to the pathogenicity/virulence of other plant fungal pathogens has also been determined, but only CK1 and PPIases show a significantly higher transcription level during pathogenesis; more detailed information associated with the pathogenicity/virulence of *P. macdonaldii* needs to be revealed.

In this study, the whole genome of *P. macdonaldii* CXJ0811 was de novo assembly based on Illumina and PacBio sequencing data that will provide a clearer picture of this fungus and the theoretical support for the pathogenesis factors against sunflower. The transcriptional analysis of the genome will focus on the genes associated with lignocellulose degrading, secondary metabolism regulation, phytotoxins biosynthesis, and transportation. The genes expressed from *P. macdonaldii* and *P. macdonaldii* on sunflowers were used to identify the genes that are potentially important during the necrotrophic phase of the disease cycle. The results presented here are the first whole genome analyses of the virulence genes associated with black stem of *P. macdonaldii* on sunflowers. These new insights into the *P. macdonaldii* genome, transcriptome, and differential gene regulation provide a better understanding of the plant-pathogen interactions that occur during black stem.

## 2. Materials and Methods

### 2.1. Fungal Strain and Growth Conditions

*P. macdonaldii* CXJ0811 was isolated from a diseased sunflower plant in Yili Prefecture, Xinjiang Autonomous Region, P. R. China. It was grown and maintained on potato dextrose agar (PDA) at 28 °C. For genomic DNA isolation, two agar plaques containing the fungal hyphae were inoculated in 100 mL of potato dextrose broth (PDB) medium, and shaken at 28 °C and 200 rpm for 7 days. For preparation of conidial suspension, 5 mm of agar plaques containing the fungal hyphae were inoculated in PDA plates and incubated for 12 days at 28 °C under continuous illumination (1500–2000 lx). For RNA isolation, approximately (1–2) × 10^6^ conidia, collected from a single plate, were used to inoculate 100 mL of PDB medium, and shaken at 28 °C and 200 rpm for 36 h. The resulting mycelial suspension was then subjected to infect sunflowers or inoculated in fresh PDB medium.

### 2.2. Phylogenetic Analysis

A phylogenetic analysis was conducted using the 18S rDNA sequences from *P. macdonaldii* CXJ0811, 21 other common plant pathogenic fungi, or 3 model fungi. The 18S rDNA sequences of other fungal species that were used in these analyses were retrieved from GenBank using the Entrez search engine in NCBI (National Center for Biotechnology Information) or DFVF (Database of Fungal Virulence Factors). These included (with GenBank accession numbers in parentheses) *Phoma macrostoma* (AB454216.1), *Phoma destructive* (AB454203.1), *Phoma herbarum* (AY293791.1), *Plasmopara halstedii* (XM_024716021.1), *Puccinia helianthi* (KM096426.1), *Golovinomyces cichoracearum* (AB769449.1), *Macrophomina phaseolina* (EF570502.1), *Sclerotinia sclerotiorum* (KC311494.1), *Botrytis cinerea* (KT587323.1), *Botryotinia fuckeliana* (JX875915.1), *Leptosphaeria biglobosa* (JF740102.1), *Alternariaster helianthi* (KC584627.1), *Alternaria alternate* (HM165489.1), *Fusarium oxysporum* (AB110910.1), *Colletotrichum graminicola* (XR_001139493.1), *Nectria haematococca* (AY489697.1), *Leptosphaeria maculans* (U04238.1), *Pyrenochaeta nobilis* (NG_062727.1), *Aspergillus niger* (DQ915806.1), *Chaetomium globosum* (AB048285.1), *Penicillium chrysogenum* (KU559908.1), *Aspergillus nidulans* (NG_064803.1), *Neurospora crassa* (XR_898131.1), and *Saccharomyces cerevisiae* (NG_063315.1). The tree was constructed with the neighbor-joining algorithm of the MEGA (version 11.0) software. The alignment gaps and missing data sites were deleted and a bootstrap value based on 1000 replications was used to measure statistical confidence of branch nodes of the phylogenetic tree.

### 2.3. Genome Sequencing and Assembly

The sequencing and assembly of *P. macdonaldii* CXJ0811 genome were carried out by a commercial provider (Azenta Life Sciences, Beijing, China). High-quality genomic DNA was extracted from 7-day-old mycelia grown in PDB, using the NucleoBond^®^ HMW DNA kit (MACHEREY-NAGEL, Düren, Germany). The genome of the CXJ0811 isolate was sequenced using both a whole-genome shotgun strategy by Illumina HiseqXten/Novaseq/MGI2000 System (paired-end, 300–350 bp) and a PacBio single-molecule real-time DNA sequencing technology [19] by Sequel II sequencing platform (10–15 kb), with a sequencing depth of 176 for the Illumina platform and >400 for the Sequel II sequencing platform.

The PacBio reads were assembled using HGAP4/Falcon (version 0.3)/Canu (version 0.7) tools [20], and the assembled results were verified using Pilon (version 1.22) software [21] with Illumina fastq data. The sequences were deposited in the GenBank database and used as queries for BLAST (version 2.2.31+) searches of the database.

### 2.4. Genomic Prediction and Genome Annotation

Gene prediction was performed using Augustus (version 3.3) [22]. Transfer RNAs (tRNAs) were detected in the genome using the program tRNAscan-SE (version 1.3.1) [23] with default parameter settings. rRNAs were identified by using Barrnap (version 0.9). Other RNAs were identified by RNA families (Rfam) database (version 12.0).

The predictive genes protein sequences were equivalenced and functional annotations were assigned to the genes using functional databases, including Clusters of Orthologous Groups (COG/KOG) [24], Gene Ontology (GO) [25], Kyoto Encyclopedia of Genes and Genomes (KEGG) [26], NCBI Non-Redundant (NR) Protein Sequence database, Swiss-Prot database, and Pfam database. Carbohydrate-active enzymes (CAZymes) were expected using the Carbohydrate-Active Enzymes database (CAZy) [27]. Secretion protein and secondary metabolite gene clusters were predicted by using the SiganalP and antiSMASH (fungal version 6.1.1) software [28], respectively. Furthermore, Pathogen-Host Interactions database (PHI) [29] and Fungal Virulence Factor database (DFVF) [30] were utilized to examine and confirm genes associated with the pathogenicity of *P. macdonaldii* CXJ0811. All BLAST alignments as mentioned above were conducted with an *E*-values threshold of ≤1 × 10^−5^. 

### 2.5. Pathogenicity Assay

Four-week-old sunflower seedlings were selected for pathogenicity assay. Roots and damaged leaves were first removed and the remaining were washed under running water. Intact leaves and stems were separated with a sterile scissor, and the stems were further cut into 5 cm segments. These tissues were washed with sterile water and subjected to disinfection: immersed in sodium hypochlorite solution (6–14% active chlorine basis) for 30 s and then washed with sterile water three times, repeating these steps twice. 

The pretreated leaves and stems were immersed in mycelial suspension for 30 s, and then were transferred to a 9 cm glass culture dish covered with sterile filter paper, respectively. The inoculated plates were transferred to a growth chamber with 50% humidity and continuous illumination (1500–2000 lx). The inoculated leaves were sampled at 2- and 6-days post-inoculation and the inoculated stems were sampled at 6-days post-inoculation. For each experimental group, at least three plates were examined and collected. For the control group, 2 mL of mycelia suspension was inoculated into 100 mL PDB medium and incubated for 2 days in a rotary shaker at 28 °C and 200 rpm. The mycelial pellets were collected by filtration. All samples were frozen in liquid nitrogen and stored at −80 °C for subsequent RNA-seq analysis.

### 2.6. mRNA Library Constructing and Sequencing

Total RNA was extracted from the lyophilized and ground 2-day-old infected sunflower leaves or 2-day-old mycelium of *P. macdonaldii* CXJ0811 using a Quick RNA Isolation Kit (Huayueyang Bioteck, Beijing, China). Illumina HiSeq sequencing of total mRNA from the *P. macdonaldii* or infected leaves and stems of the sunflower was conducted by Azenta Life Sciences (Beijing, China; https://www.genewiz.com, accessed on 20 October 2022). The 1 μg total RNA was used for the following library preparation. The poly (A) mRNA isolation was performed using Oligo (dT) beads. The mRNA fragmentation was performed using divalent cations and high temperatures. Priming was performed using Random Primers. First-strand cDNA and the second-strand cDNA were synthesized. The purified double-stranded cDNA was then treated to repair both ends and add a dA-tailing in one reaction, followed by a T-A ligation to add adaptors to both ends. Size selection of Adaptor-ligated DNA was then performed using DNA Clean Beads. Each sample was then amplified by PCR using P5 and P7 primers and the PCR products were validated. Then, libraries with different indexes were multiplexed and loaded on an Illumina HiSeq/Illumina Novaseq/MGI2000 instrument for sequencing using a 2 × 150 paired-end (PE) configuration according to manufacturer’s instructions.

### 2.7. Data Mining and Differential Expression Analysis

Raw reads were processed using Cutadapt (version 1.9.1) to remove technical sequences. The resulting clean reads were mapped to the *P. macdonaldii* reference genome (CXJ0811) using HISAT2 (version 2.1.0) [31]. Gene expression values were calculated as fragments per kilobase of transcript per million mapped reads (FPKM) using HTSeq (version 0.6.1) [32]. Differential expression analysis used the DESeq2 Bioconductor package [33], a model based on the negative binomial distribution. *p*-Values were used to evaluate expression differences at a statistically significant level. A false discovery rate (FDR)-corrected *p*-value ≤ 0.05 and absolute value of log 2 Ratio ≥ 1 were used to identify the differentially expressed genes (DEGs) and differentially expression tags (DETs).

### 2.8. RNA Preparation and Quantitative Real-Time (qRT)-PCR 

Total RNA was extracted from the lyophilized and ground mycelium using a TRIzoL kit (Invitrogen, CA, USA). The quantity and quality of the total RNA were detected by a NanoDrop^®^ ND-2000 spectrophotometer (Thermo Scientific, Shanghai, China). The first-strand cDNA was generated by reverse transcription in a 20 μL reaction using HiScript III RT SuperMix for qPCR kit (Vazyme, Nanjing, China). QRT-PCR was performed by Applied Biosystems QuantStudio^TM^5 (Thermo Scientific, Shanghai, China). Each reaction of 20 μL PCR was performed with PowerUp SYBR Green Master Mix (Thermo Scientific, Shanghai, China). Reactions were set up in three replicates per sample. Controls without addition of the templates were included for each primer set. PCR cycling parameters were the following: pre-incubation at 50 °C for 2 min and 95 °C for 5 min, followed by 40 cycles of denaturation at 95 °C for 15 s, and annealing and extension at 60 °C for 1 min. qRT-PCR data were analyzed using the 2^−ΔΔCt^ relative quantification method [34] to calculate relative expression levels of genes. The housekeeping genes encoding actin (Actin) served as a reference. The amplification efficiencies of the target and reference genes were compared at different template concentrations. The gene-specific pairs of primers used in the amplifications were listed in Appendix A. 

## 3. Results

### 3.1. Phylogenic Analysis of P. macdonaldii CXJ0811

We constructed a phylogenetic tree with 18S rDNA sequences from *P. macdonaldii* CXJ0811 and 24 other fungal strains, including 21 plant pathogenic fungi and three model fungi to demonstrate its taxonomic status (Figure 1). The *P. macdonaldii* CXJ0811 was clustered together with *Phoma lingam* and *Phoma macrostoma* with 87.61% and 71% similarity, respectively. It is suggested that CXJ0811 belongs to the genus of the *Phoma* pathogenic fungi.

### 3.2. Genome Structures

The predictions and the statistics of the genome are summarized in Table 1 and Table 2. The draft genome sequence of *P. macdonaldii* CXJ0811 was 38.24 Mb and consisted of 27 contigs with an N50 of 1508 kb and a G + C content of 48.27%. A total of 11,094 genes were predicted by Augustus (version 3.3); 45 tRNA genes and 73 rRNA genes were predicted by tRNAscan-SE and Barrnap, respectively. The genome sequence of *P. macdonaldii* CXJ0811 has been deposited in GenBank under the accession number GCA_023566075.1 (BioProject: PRJNA836725; BioSample: SAMN28167593. The predicted protein and annotations are shown in Appendix A).

### 3.3. Gene Annotation

A total of 10,933 protein-encoded genes were annotated in the assembled genome (Table 2). KOG, GO, KEGG, Swiss-Prot, and NCBI nr were performed to annotate the gene function of *P. macdonaldii* CXJ0811.

In the KOG analysis, 5564 (50.15%) genes were redundantly assigned into KOG classifications (Figure 2), of which 1027 genes were in the category of general function prediction only [R], 528 genes were annotated to post-translational modification, protein turnover, and chaperones [O], 410 genes were annotated to signal transduction mechanisms [T], 345 genes were annotated to secondary metabolites biosynthesis, transport, and catabolism [Q], 289 genes were annotated to transcription [K], 326 genes were annotated to carbohydrate transport and metabolism [G], and 297 genes were annotated to amino acid transport and metabolism [E]. 

In the GO analysis, 5696 (50.09%) of 10,933 predicted genes were given a GO assignment (Figure 3). Of these genes, 6672 were redundantly assigned to molecular function ontology, 4795 to biological process ontology, and 2513 to cellular component ontology. Most of the genes were annotated to binding activity (3018 genes) and catalytic activity (2816 genes) in the molecular function ontology classification, metabolic process (2221 genes), cellular process (1196 genes), and localization (727 genes) in the biological process ontology classification, and membrane (566 genes) and organelle (477 genes) in the cellular component ontology classification. 

To further understand the functions of *P. macdonaldii* CXJ0811 proteins, 2224 (20.34%) genes were assigned to their orthologs in the KEGG database and could be divided into six types or 46 categories based on their functions (Figure 4 and Appendix A). A total of 1524 genes were in the metabolism type that accounted for the largest proportion; this is consistent with the results of the biological pathway function annotation in GO. Most genes were mapped to the signal transduction (426 genes), carbohydrate metabolism (318 genes), amino acid metabolism (267 genes), and transport and catabolism (256 genes) categories, 2018 (18.46%) genes were mapped to the KEGG database, and this is consistent with the KOG feature annotation results (Figure 2 and Figure 4). A total of 60 genes were mapped to the xenobiotics biodegradation and metabolism category, 47 genes were mapped to the category of biosynthesis of other secondary metabolites, and 29 genes were mapped to the category of metabolism of terpenoids and polyketides. A total of 13 enzymes were mapped in the terpenoid backbone biosynthesis, 15 enzymes were mapped in the phenylalanine, tyrosine, and tryptophan biosynthesis, precursors of polyketides [35], 7 enzymes were mapped in the streptomycin biosynthesis, 15 enzymes were mapped in the MFS transporters, and 9 enzymes were putative ABC transporters. The fungal secondary metabolites, such as terpenoids and polyketides, as well as their transporters, were essential for the pathogenesis of the pathogenic fungi [36,37].

A total of 7027 genes (64.27%) had similarities to the proteins in the Swiss-Prot database, 7598 (69.49%) were mapped to the Pfam database, and 10,205 (93.34%) were mapped to the NCBI nr database, which shared a 59.44% similarity with *Leptosphaeria maculans* (anamorph: *P. lingam*) (Table 2 and Figure 5). 

The gene annotation information indicated that there were 5564, 5696, 2224, 7027, and 10,205 annotated genes in the KOG, GO, KEGG, Swiss-Prot, and NCBI nr databases. The total number of annotated genes was 10,933, and the total number of unannotated genes was 161.

### 3.4. Genes Involved in Carbohydrate Degradation (CAZymes)

For plant pathogens, CAZymes are critical in the process of plant pathogen infection, especially in the cell wall structural components and the metabolism of stored glucan [38]. The analysis of CAZymes showed that *P. macdonaldii* CXJ0811 contained CAZymes specific for plant polysaccharide degradation, with a total of 1133 putative functional domains. The modules comprise 384 domains belonging to glycoside hydrolases (GH), 303 to glycosyltransferases (GT), 183 to carbohydrate-binding modules (CBM), 157 to auxiliary activities (AA), 80 to carbohydrate esterases (CE), and 26 to polysaccharide lyases (PL) (Figure 6a). Based on substrate specificity, *P. macdonaldii* CXJ0811 has a high number of CAZymes involved in the degradation of cellulose (40 genes), hemicellulose (47 genes), and pectin (46 genes), the main components of a plant’s cell wall (Appendix A). It also harbors a larger number of CAZyme modules involved in starch (46 genes) degradation, indicating a possible preference of this fungus toward the storage organs of plants, such as seeds, roots, and tubers. Some modules, such as endo-1,3-beta-glucosidase (GH17), chitinase (GH18 and GH19), and chitosanase (GH5 and GH75), involved in fungal cell wall degradation, and lysozyme (GH23), involved in bacterial cell wall degradation, were also abundant in this isolate. This might endow the antagonistic advantage of *P. macdonaldii* CXJ0811 against other microorganisms [35,39].

### 3.5. Pathogenesis Related Genes

We screened the *P. macdonaldii* CXJ0811 genome sequence against PHI, a database that collects pathogenicity, virulence, and effector genes from fungi, oomycetes, and bacterial pathogens. A total of 2356 (21.24%) of the *P. macdonaldii* CXJ0811 genes were predicted to be involved in an interaction between the pathogen and host, the number was more than that of *P. lycopersici* (2196 genes) [40], but less than *P. sorghina* var. saccharum (2440 genes) [41], and *P. arachidicola* (3415 genes) [42]. About one-third (697 genes) of the predicted pathogenic genes were homologous to *Fusarium oxysporum*. The predicted pathogenic genes of *P. macdonaldii* CXJ0811 were further classified into nine categories (Figure 6b), of which most of the genes were related to the “Reduced virulence” (1116 genes), “Unaffected pathogenicity” (979 genes), or “Loss of pathogenicity” (232 genes) categories. The genes belonging to the “Increased_virulence_ (hypervirulence)” category are the key pathogenic genes, and 107 genes were mapped to this phenotype, with annotations as cytochromeP450 family, major facilitator superfamily (MFS) transporter, ATP-dependent RNA helicase, sensory transduction histidine kinase, GTP-binding protein, WD40 repeat-like protein, scytalone dehydratase, transcription factor, and pectate lyase (Appendix A). The proteins belonging to the cytochrome P450 or MFS family are generally believed to play important roles in the biosynthesis and transportation of metabolites. The GTP-binding protein and sensory transduction histidine kinase are involved in cellular signal transduction as binary molecular switches. The WD40 proteins are associated with numerous biological processes, including the conidiation, growth, and pathogenicity of fungi [43]. The ATP-dependent RNA helicases play roles in the recognition of foreign nucleic acids and the modulation of viral infection [44]. In addition, 15 genes belonged to the “Effector” category, and one gene in the “Enhanced antagonism” category, these genes were also associated with the pathogenesis of *P. macdonaldii* CXJ0811.

Although PHI-base collected pathogenic genes for all types of fungal and bacterial pathogens, it contained a limited number of fungal virulence factors, and most of its records are about bacterial pathogens [30]. Thus, a genome-wide BLAST analysis against the DFVF was performed, and 2167 genes were predicted to be associated with the virulence factors of *P. macdonaldii* CXJ0811. A total of 1070 genes (49.37%) were involved in herb, plant, or xyloid diseases, such as the southern leaf blight of maize, witches’ broom, corn smut, seedling blight, rice blast, powdery mildew, leaf spot, the late blight of potato and tomato, the gray leaf spot of corn, glume blotch, blight, canker, and anthracnose (Appendix A). Most of these genes were associated with leaf spot (343) and rice blast (134). The BLAST analyses against the PHI database indicated that 685 (64.01%) of these genes were putatively involved in pathogen-host interactions. Similar to the PHI analysis results, the majority of these plant virulence genes were involved in plant cell wall degrading, metabolites biosynthesis and transportation, signal transduction, and transcription regulation. These predicted fungal pathogenic factors associated with plant diseases were orthologous to 43 other pathogenic fungi. Among them, 297 genes (27.71%) were orthologous to *Magnaporthe* sp., 164 genes were orthologous to *Botryotinia fuckeliana* (15.30%), 87 genes were orthologous to *Fusarium* sp. (8.12%), 84 genes were orthologous to *Alternaria* sp. (7.84%), 75 genes were orthologous to *Ustilago maydis* (7.00%), and 72 genes were orthologous to *Phaeosphaeria nodorum* (6.72%) (Figure 6c).

Plant fungi can produce diverse secondary metabolites (SMs) that aid pathogenicity [15,45]. Thus, the gene clusters for the secondary metabolites of *P. macdonaldii* CXJ0811 were predicted by antiSMASH. There were 162 genes and 12 clusters in Type I polyketide synthase (T1PKS), 140 genes and 12 clusters encoding nonribosomal peptide synthetases (NRPS), 84 genes and six clusters in NRPS-T1PKS (indole were noticed), and 55 genes and seven clusters in terpene (Figure 6d and Appendix A). All of these gene clusters were predicted to be involved in the pathogenesis of *P. macdonaldii* CXJ0811, of which 138 genes (31.29%) were mapped to the PHI database and 139 genes (31.51%) were mapped to DFVF (Appendix A). 

Secretory proteins play crucial roles during the early infection of pathogenic fungi. A total of 1057 signal peptide protein structures, 2181 proteins with transmembrane structures, and 827 secreted proteins were predicted in the genome of *P. macdonaldii* CXJ0811 (Table 2). The putative secretory proteins were annotated using the PHI database and were further searched against the DFVF database. The BLAST analyses indicated that 202 of the secretory proteins in *P. macdonaldii* CXJ0811 (accounting for 24.42% of the total secretome) were putatively involved in pathogen-host interactions and 223 (26.96%) were annotated as fungal virulence factors (Appendix A). 

### 3.6. Transcriptomic Pattern of Genes from Different Diseased Tissues

In order to verify the predicted pathogenesis-related genes that screened from the genome analysis, we utilized a replicate set of infected leaves and stems of sunflowers to obtain transcriptomes representing the necrotrophic phases of the disease. As shown in Figure 7a, the *P. macdonaldii* CXJ0811-inoculated leaves and stems showed the first signs of colonization after culture for 2 days while, at 6 days, the percentage of the colonized fungal spots was about 100%. The control groups did not show any fungal spots throughout the experiment. However, the lower fungal abundance observed in the stem samples inoculated after 2 days of culture may lead to insufficient sequencing coverage in the samples. Based on these data, the RNA-Seq analyses were performed on 2-day infected leaves (LEAF-2d), 6-day infected leaves (LEAF-6d), 6-day infected stems (STEM), and 2-day mycelium of *P. macdonaldii* CXJ0811 (CT). The samples of CT, LEAF-2d, LEAF-6d, and STEM yielded 43419308, 37209286, 78,841,080, and 34,809,958 high-quality fungal reads, respectively. After aligning the high-quality reads to the genome sequence of *P. macdonaldii* CXJ0811, the percentage of the mapped reads of each sample to the genome sequence ranged from 0.49% to 95.39% (Appendix A).

The DEGs between the *P. macdonaldii* CXJ0811 and infected sunflower tissues were selected based on the FDR-corrected *p*-value ≤ 0.05 and the absolute value of log 2 Ratio ≥ 1. A total of 2506 DEGs were found in LEAF-2d, of which 1357 genes were up-regulated and 1149 genes were down-regulated. A total of 3035 DEGs were found in LEAF-6d, of which 786 genes were up-regulated and 2249 genes were down-regulated. A total of 2660 DEGs were found in STEM, of which 1444 genes were up-regulated and 1160 genes were downregulated (Figure 7b). 

As the fungal virulence factors were probably up-regulated during the infection process [46], more attention was paid to the genes up-regulated in these samples. In the GO analysis (Figure 8), the up-regulated genes of these diseased sunflower tissues were uniformly enriched in the GO-term of the extracellular region (GO: 0005576). In sample LEAF-2d, the up-regulated genes were also enriched in GO-terms of the structural constituent of ribosome (GO: 0003735) and translation (GO: 0006412), suggesting a busy protein biosynthesis process occurred during the early stages of infection. With the extension of the infection time, more actions were focused on the transportation and activation of mature proteins, thus, the secondary significant GO-terms were cytoplasm (GO: 0005737) and metal ion binding (GO: 0046872) in sample LEAF-6d. In order to degrade the rigid cell wall of the stems, the genes involved in the cellulose catabolic process (GO: 0030245), xylan catabolic process (GO: 0045493), cell wall organization (GO: 0071555), and pectate lyase activity (GO: 0030570) were also up-regulated. In the KEGG analysis (Figure 9), the most significant pathways of the DEGs from these diseased sunflower tissues were the metabolic pathways and biosynthesis of secondary metabolites. The majority of the up-regulated genes encoded with enzymes were also involved in these pathways.

### 3.7. Transcriptomic Pattern of Pathogenesis Genes

The analysis of the DEG intersections showed that there were 751 DEGs with 371 up-regulated and 359 down-regulated genes in common among LEAF-2d, LEAF-6d, and STEM (Appendix A). As our present focus is on the pathogenesis genes, these up-regulated genes were subjected to further BLAST analysis in Appendix A to confirm previous predictions based on genome analysis. Among these up-regulated genes, 82 (22.10%) were mapped to DFVF and 31 were associated with leaf spot; 63 (16.98%) were mapped to PHI-base and predicted to be involved in pathogen-host interactions; 69 (18.59%) were annotated as CAZymes that were involved in cellulose, hemicellulose, pectin, and chitin degrading; 33 (8.89%) were annotated as transporters; and 91 (24.52%) were annotated as secretory proteins. 

A hierarchical cluster analysis of the altered genes located in the secondary biosynthesis gene clusters from our RNA-seq analysis further defined the important secondary metabolites involved in the pathogenesis of *P. macdonaldii* to sunflowers (Figure 10a). A total of 49 genes (including three carbon skeleton biosynthetic genes) distributed in 21 gene clusters were up-regulated in LEAF-2d, 39 genes (including three carbon skeleton biosynthetic genes) distributed in 24 gene clusters were up-regulated in LEAF-6d, and 57 genes (including seven carbon skeleton biosynthetic genes) distributed in 26 gene clusters were up-regulated in STEM. A total of 35 up-regulated genes were shared among these samples, of which a carbon skeleton biosynthetic gene (Pm_7268 of cluster 23) was found. Pm_7268, predicted to encode an NRPS-like enzyme, showed different degrees of up-regulation in LEAF-2d, LEAF-6d, and STEM, with LogFC values of 5.91, 5.01, and 3.28, respectively. It is suggested that the secondary metabolite synthesized by gene cluster 23 was probably the main virulence effector in the pathogenesis of *P. macdonaldii* to sunflowers.

To validate the accuracy of the RNA-seq and confirm the key role of the secondary metabolites in fungal pathogenesis, RT-qPCR was performed. A total of 15 genes with high gene expression variation (logFC > 5.00), including seven MFS transporters, a pectate lyase, a beta-glucosidase, a Na+-exporting ATPase, a rhamnogalacturonate lyase, an NRPS-like enzyme, a metallocarboxypeptidase, an oxidoreductase, and a hypothetical protein, were selected from all the DEGs. The primer sequence is shown in Appendix A. Consistent with the transcriptome profiling analysis, all the genes were significantly up-regulated in the LEAF-2d sample, ranging from 21- to 2718-fold of the control (Figure 10b), confirming the accuracy of the transcriptome profiling obtained with RNA-seq and provided further evidence of the importance of these genes in *P. macdonaldii* pathogenesis. It should be noted that MFS Pm_8157 and Pm_369 had the highest variation of mRNA levels, with approximately 2718- and 1427-fold of the control, respectively. 

## 4. Discussion

A previous study on sunflower black stem mainly focused on disease-resistant sunflower lines screening [47] and resistance gene identification [13,14,48]; little attention has been paid to the molecular characteristics of the causal agent *P. macdonaldii*, and the genomic information, and the genes involved in the pathogenesis of this pathogen is still unclear. In this study, we completed the draft genome sequence of *P. macdonaldii* CXJ0811 using both the whole-genome shotgun strategy and PacBio single-molecule real-time DNA sequencing technology, with high-quality and completeness. The genome annotation revealed *P. macdonaldii* has an extensive array of CAZymes for breaking down the plant cell wall, a considerably diverse set of genes involved in extracellular perception and signal transduction, and phytotoxic secondary metabolites for disturbing plant immune responses. A further transcription profiling analysis revealed 82 DFVF genes, 63 PHI genes, 69 CAZymes, 33 transporters, 91 secretory proteins, and putative host-specific toxins (HSTs) biosynthesized by gene cluster 23 were significantly up-regulated as *P. macdonaldii* attacked and colonized the sunflower. This is the first analysis of the genome and transcriptome of *P. macdonaldii*, which allowed a comprehensive understanding of the likely attributes required by the plant pathogenic fungus *P. macdonaldii* to invade and colonize sunflowers.

### 4.1. Role of Plant Cell Wall-Degrading Enzymes (CWDEs) in Pathogenesis of P. macdonaldii 

The fungal pathogen entry into plants is an essential step that determines the success of infection and disease development. Previous reports demonstrated that *P. macdonaldii* penetrated into the plants either directly via the enzymatic degradation of the plant cell wall or by mechanical pressure, or indirectly through wounds and natural openings, such as lenticels and stomata [49,50]. A CAZymes analysis (Appendix A) revealed a high number of plant cell wall-degrading enzymes (CWDEs), including xyloglucanases, polygalacturonases, glucanases, cellulases, and pectinases that were secreted by *P. macdonaldii*, signifying the enzymatic strategy for *P. macdonaldii* infection. Further confirmation of the CWDE importance for *P. macdonaldii* virulence was obtained from transcriptome analysis. As shown in Figure 8, the genes involved in the cellulose catabolic process (GO: 0030245), xylan catabolic process (GO: 0045493), and pectate lyase activity (GO: 0030570) were uniformly up-regulated either in infected leaves or stems. In order to degrade the rigid cell wall of the stems, three genes (Pm_2003, Pm_3131, and Pm_8541) involved in the lignin catabolic process (GO: 0046274) were also up-regulated, with LogFC values of 7.43, 4.86, and 7.83, respectively. Previously, it was generally accepted that the effect of CWDEs on pathogens’ virulence was dependent on their enzymatic activity. However, several studies have shown that at least some CWDEs may also act as virulence factors, either dependent or independent of their enzyme degradation activity [51]. Several studies demonstrated that multiple CWDEs are detected as pathogen-associated molecular patterns (PAMPs) by plant innate immune systems to induce responses [52,53]. The deficiency in CWDEs led to a substantial loss of virulence in plants [54,55].

### 4.2. Role of Effectors in Pathogenesis of P. macdonaldii

To facilitate infection, plant pathogens are known to secrete effectors directly into host plant cells to suppress ROS generation, perturb host cell signaling, inhibit the plant’s innate immune system, or interfere with the biosynthesis of hormones and secondary metabolites [51]. The plant adaptive immune system has, in turn, evolved to recognize pathogen effector proteins by intracellular immune receptors (which are historically referred to as resistance (R) proteins and encoded by *R* genes) [56]. Effector-specific recognition happens according to the gene-for-gene concept, in which each effector gene in the pathogen has a counterpart major *R* gene in the plant [57]. 

*R*-*Avr* (avirulence effector proteins) gene interactions are frequently exploited in the field to control diseases. Although it has been recognized that significant differences in pathogenicity exist among different *P. macdonaldii* isolates on the same genetic material [17], *Avr* genes of *P. macdonaldii*, and *Avr* profiles of *P. macdonaldii* populations, are still unknown. Therefore, the major *R* genes of the sunflower response to *P. macdonaldii* are still undetermined. In this present study, 15 genes were annotated as effectors (plant avirulence determinants) (Appendix A). The sequence alignment with *P. lingam* AVR proteins obtained two homologous genes: Pm_8184 and Pm_2515, which share 46.67% and 31.71% similarity with AVRLm2 of *P. lingam*, respectively. A total of 12 of these genes were annotated as CAZymes, five were mapped to DFVF and annotated as fungal virulence factors, 10 were predicted secretory proteins, and three were located in secondary metabolite gene clusters. RNA-seq analysis results further revealed that Pm_9144, Pm_9244, Pm_9146, and Pm_8618 were uniformly significantly up-regulated either in LEAF-2d or LEAF-6d as compared with CT. In STEM-6d, five genes were significantly up-regulated, including Pm_9144, Pm_8618, and another three genes. It is suggested that different combinations of *Avr* genes are involved in different tissue infections, and these up-regulated genes are considered candidates of sunflower-specific *Avr* genes for *P. macdonaldii*. 

Necrotrophic effectors were also searched in the genome of *P. macdonaldii* by protein BLAST, and the amino acid sequences of the necrotrophic effectors reviewed by Doehlemann et al. [51] were used as references. In total, six homologous genes (Pm_123, Pm_178, Pm_393, Pm_2266, Pm_2291, and Pm_2294) were obtained. Pm_178, annotated as a secretory protein and mapped to the “Increased Virulence_ (Hypervirulence)” category of PHI, encoded a BcSpl1-like protein (with a 54.10% similarity), which was previously reported to enhance host susceptibility to *Botrytis cinerea* [58]. Pm_123 and Pm_2266 were homologous genes of Ss-Caf1, which was reported to trigger host cell death during the early stages of *Sclerotinia sclerotiorum* infection and play a significant role in the formation of infection cushion and sclerotial development [59]. Pm_393 encoded an SsCm1-like protein with a 36.80% similarity, which was previously reported to have a high structural and functional similarity to the *U. maydis* effector Cmu1 [60]. However, the homologous gene of Cmu1 was Pm_2037 (with a 30.31% similarity) in *P. macdonaldii*. Pm_2291 (with a 40.62% similarity) encoded an SsCVNH-like protein, which was previously reported to be a cysteine-rich secretory protein and is essential for the virulence and sclerotial development of *S. sclerotiorum* [61]. The protein encoded by Pm_2294 shared 35.82% and 37.50% similarity with *S. sclerotiorum* necrotrophic effectors SsNep1 and SsNep2, respectively [62]. Three of these genes were significantly differentially expressed in infected sunflower tissues: Pm_178 was down-regulated in LEAF-6d, with a LogFC value of −3.96; Pm_2266 was down-regulated both in LEAF-6d and STEM, with LogFC values of −12.33 and −2.05, respectively; while Pm_2291 was up-regulated both in LEAF-2d and STEM, with LogFC values of 2.66 and 1.97, respectively. It is suggested that the protein encoded by Pm_2291 probably plays an important role in *P. macdonaldii* infection.

### 4.3. Role of Phytotoxins in Pathogenesis of P. macdonaldii

Once successfully infected, under favorable environmental conditions, fungal pathogens can secrete a variety of plant toxic secondary metabolites, also called phytotoxins, which induce cell death and contribute to the development of plant disease symptoms [45,63,64]. As a famous plant pathogen, the secondary metabolites produced by the genus of *Phoma* have attracted extensive attention, and various phytotoxins have been revealed from *P. tracheiphila* [64], *P. clematidina* [65], *P. betae* [66], *P. herbarum* [67], *P. chenopodiicola* [68], *P. putaminum* [69], *P. cava* [70], *P. asparagi* Sacc [71], *P. destructiva* [72], *P. exigua var. exigua* [73], and *P. lingam* [15]. Most of these reported phytotoxins belong to aromatic polyketides and sesquiterpenoids. Although there is a dearth of information about the secondary metabolites from *P. macdonaldii*, a wealth of phytotoxins have been isolated from *P. lingam*, a close relative of *P. macdonaldii* (Figure 1) [15], which may provide some clues for the chemical structure identification of the phytotoxins from *P. macdonaldii*. Fungal phytotoxins are classified into HSTs and non-host selective (non-HSTs) categories. The reported HSTs of *P. lingam* were phomalide, phomalairdenone, and depsilairdin that function as essential determinants of pathogenicity or virulence and could cause disease symptoms similar to those caused by the pathogen, suggesting an available screening method for the resistance varieties by the selection of plant material for toxin resistance [15]. In some cases, fungal phytotoxins act as general or race-specific elicitors that are expected to induce the production of phytoalexins in cultivars. Thus, the determination of the bioactivity characteristics of the secondary metabolites in the pathogenesis process of *P. macdonaldii* is important in controlling black stem. In this study, secondary metabolite biosynthetic gene clusters 1, 2, 9, 23, and 36 were up-regulated in all sunflower-infected tissues, and key genes of cluster 23 were proved to be up-regulated to 23- to 126-fold of the control (Figure 10). The secondary metabolites synthesized by these gene clusters probably facilitate infection and the development of plant disease symptoms. The data on predicted secondary metabolites gene clusters and their corresponding expression profile in this paper will facilitate further exploration of the HSTs of *P. macdonaldii* by gene knockout, overexpression, and pathogenicity analysis. 

## 5. Conclusions

In this study, a high-quality annotated genome of *P. macdonaldii*, the causal agent of sunflower black stem, was established. Genes with encoded CAZymes, pathogenicity factors, effectors, secretory proteins, transporters, and phytotoxins were identified in the *P. macdonaldii* genome. The importance of these genes in the pathogenesis of *P. macdonaldii* was further confirmed through a comprehensive transcriptome analysis. 

The whole-genome sequencing and transcriptomic analysis of *P. macdonaldii* CXJ0811 provided important data and the theoretical basis for better understanding the genomic features of *Phoma* species, the elucidation of its pathogenesis, and supporting the prevention and control of the diseases caused by this pathogenic fungus. The repertoire of the genes up-regulated at both the early and late stages of the black stem occurred will contribute toward detailed further research about the fungal pathogenicity of sunflowers. Moreover, this study will provide valuable resources for genomic studies and reference information for the management of different *Phoma* species, as well as other related fungi. In conclusion, we provided baseline data on the significant pathogenicity characteristics within the genus at the genome level, and the putative virulence factors of *P. macdonaldii*, which may be potential targets for further research and the prevention of black stem. 

## Figures and Tables

**Figure 1 jof-09-00520-f001:**
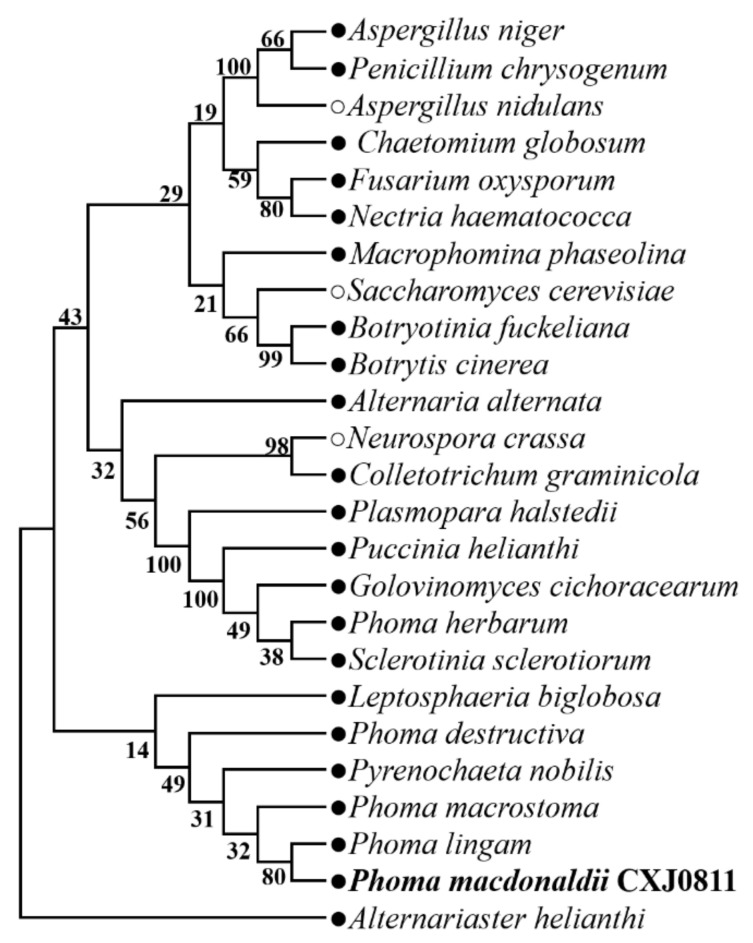
Phylogenetic relationships among common plant pathogenic fungi based on the 18S rDNA gene sequence. The tree was generated using MEGA (version 11.0). CXJ0811 was found clustered within the genus of *Phoma*. The tree was constructed according to the neighbor-joining method, with 1000 bootstrap replicates. Branch values (>70) are indicated and the scale bar represents approximately 1% nucleotide differences between close relatives.

**Figure 2 jof-09-00520-f002:**
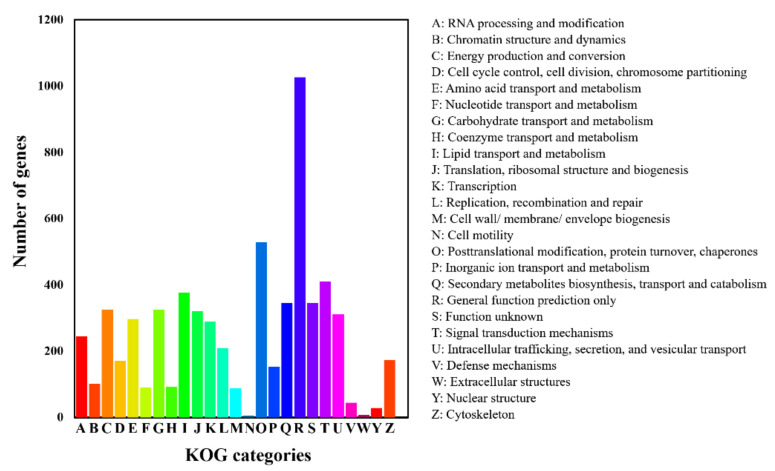
KOG classification of predicted genes in *P. macdonaldii* CXJ0811.

**Figure 3 jof-09-00520-f003:**
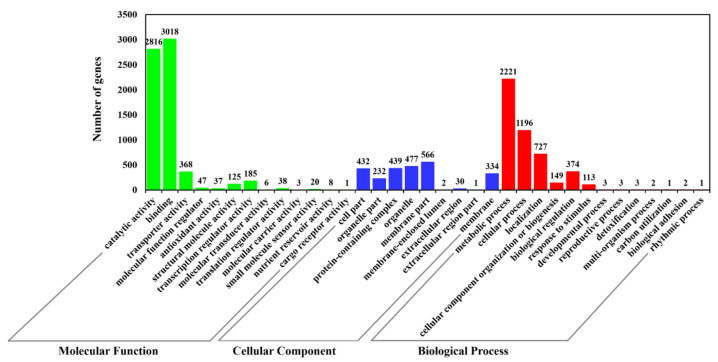
GO classification of predicted genes in *P. macdonaldii* CXJ0811.

**Figure 4 jof-09-00520-f004:**
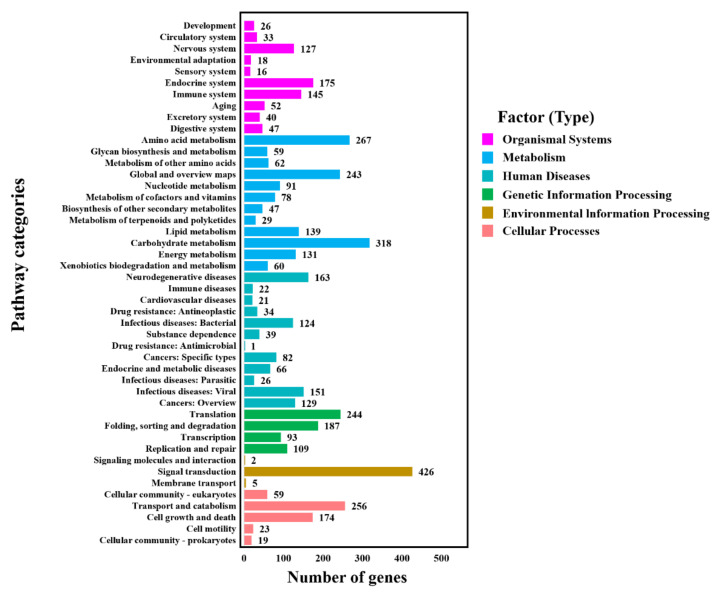
KEGG classification of predicted genes in *P. macdonaldii* CXJ0811.

**Figure 5 jof-09-00520-f005:**
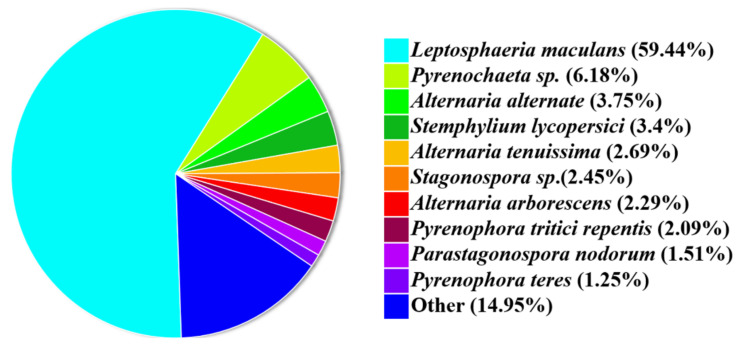
NR homologous species distribution.

**Figure 6 jof-09-00520-f006:**
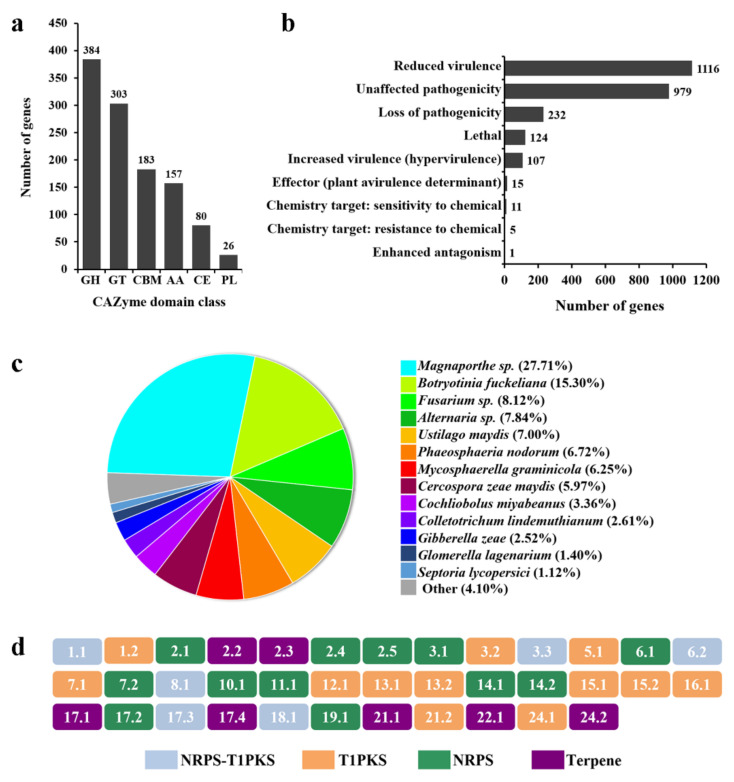
Pathogenesis-related genes in *P. macdonaldii* CXJ0811 genome. (**a**) CAZymes encoding genes, including biomass-degrading enzymes, are represented. GH: glycoside hydrolases, GT: glycosyltransferases, CBM: carbohydrate-binding modules, AA: auxiliary activities, CE: carbohydrate esterases, PL: polysaccharide lyases. (**b**) Phenotype classification of putative pathogenicity genes retrieved from the pathogen-host interactions database (PHI-base). (**c**) Putative pathogenicity genes retrieved from the database of fungal virulence factors (DFVF). (**d**) Predicted gene clusters for secondary metabolites retrieved from antiSMASH (fungal version 6.1.1).

**Figure 7 jof-09-00520-f007:**
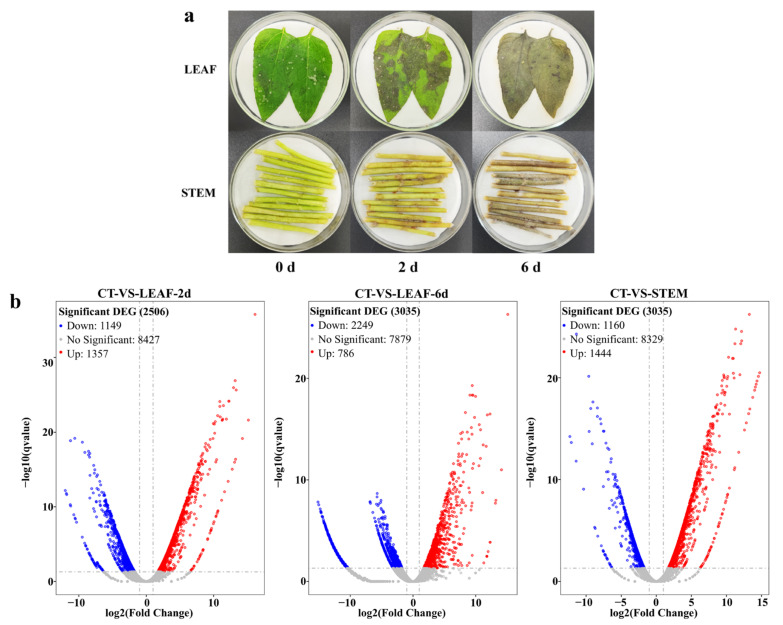
RNA-seq transcriptional profiling of *P. macdonaldii* CXJ0811 and infected sunflower tissues. (**a**) Pathogenicity bioassay of *P. macdonaldii* CXJ0811. Necrotic areas appeared on leaves and stems of sunflowers after being inoculated with 36-h-old mycelial pellets of *P. macdonaldii* CXJ0811 for 2 days. Necrotic areas were enlarged and even covered the sunflower tissues on 6th day. (**b**) Volcano maps display the significant DEGs between CT and each treatment group (LEAF-2d, LEAF-6d, and STEM). Blue dots represent genes that were significantly down-regulated; red dots represent genes that were significantly up-regulated; and grey dots represent genes for which no significant differences were observed).

**Figure 8 jof-09-00520-f008:**
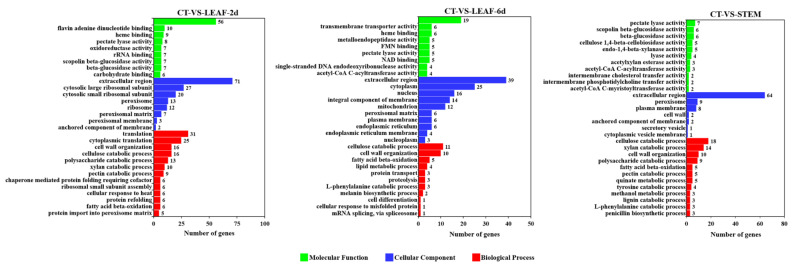
GO annotation of up-regulated genes between infected sunflower tissues (LEAF-2d, LEAF-6d, and STEM) and *P. macdonaldii* CXJ0811 (CT).

**Figure 9 jof-09-00520-f009:**
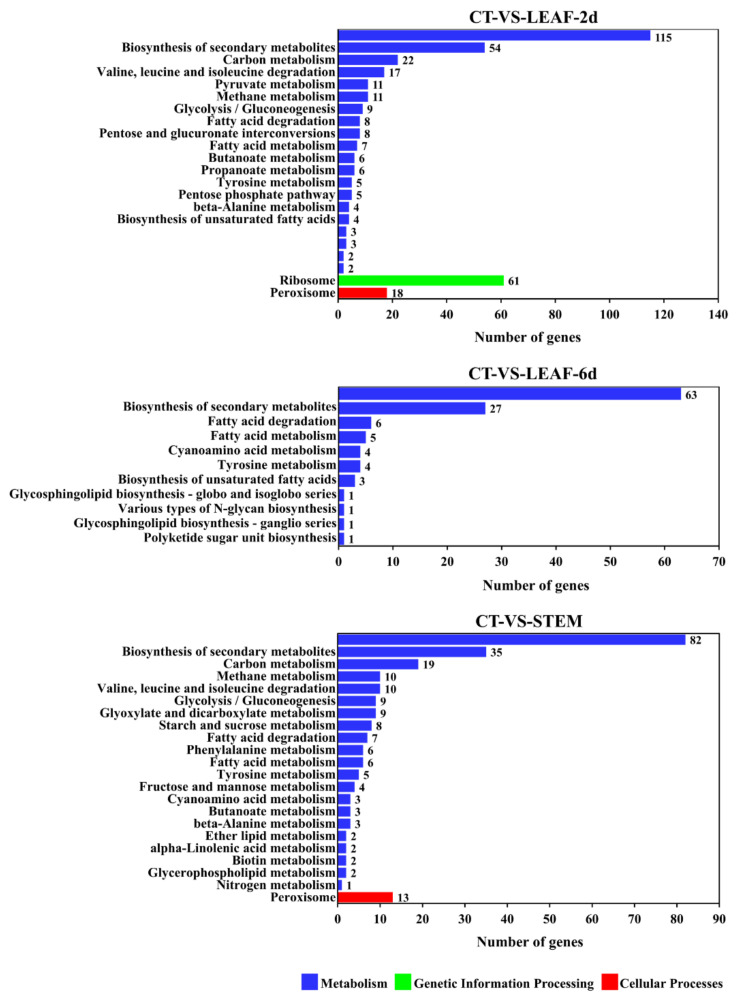
KEGG annotation of up-regulated genes between infected sunflower tissues (LEAF-2d, LEAF-6d, and STEM) and *P. macdonaldii* CXJ0811 (CT).

**Figure 10 jof-09-00520-f010:**
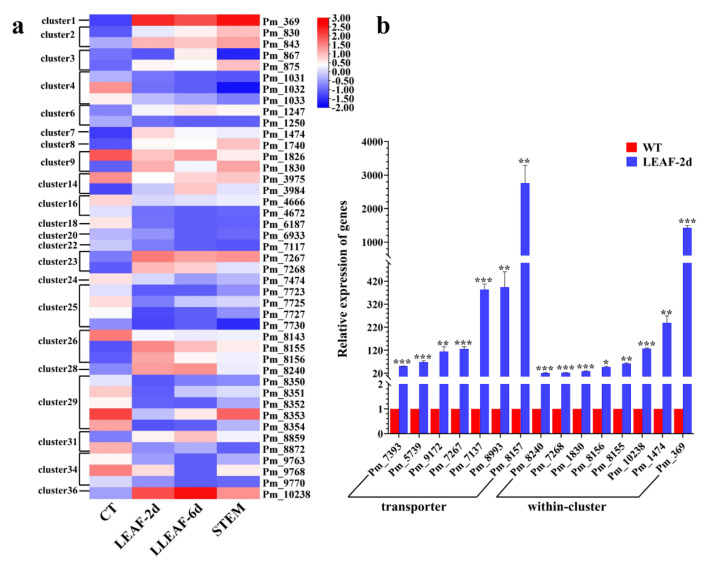
Expression of genes involved in secondary metabolite biosynthesis and transportation in *P. macdonaldii* CXJ0811 and infected sunflower tissues. (**a**) Heat map of common DEGs of secondary metabolite biosynthesis gene clusters from infected sunflower tissues (LEAF-2d, LEAF-6d, and STEM) as compared with *P. macdonaldii* CXJ0811 (CT). The log10 (FPKM + 1) value of each gene is used for clustering. Red represents highly expressed genes, and blue represents low expressed genes. Different colors represent distinct relative expressions of these genes. (**b**) Relative expression levels by qRT-PCR analysis of 15 genes with high gene expression variation, chosen from the RNA-seq profiling and annotated as transporters or found in the secondary metabolites biosynthetic gene clusters, in the sample LEAF-2d compared with CT. The 8 genes within clusters are Zn-dependent exopeptidase (Pm_8240), NRPS-like enzyme (Pm_7268), beta-glucosidase-like protein (Pm_1830), ester hydrolase (a DUF1907-domain-containing protein) (Pm_8156), NAD(P)-binding protein (Pm_8155), sodium P-type ATPase (Pm_10238), rhamnogalacturonate lyase (Pm_1474), and polysaccharide lyase family 1 protein (Pm_369), respectively. All transcripts were normalized against actin cDNA amplified with primers actin-F and actin-R (S1 Appendix). All experiments were performed in triplicate. There is a significant difference between the CT and LEAF-2d as indicated by one to three asterisks (*p*-value < 0.05 = *, *p*-value < 0.01 = **, *p*-value < 0.001 = ***, with *t*-test analysis).

**Table 1 jof-09-00520-t001:** Assembly summary.

Attribute	Value
Number of contigs	27
Total contigs size (bp)	38,245,005
Min contigs length (bp)	40,900
Max contigs length (bp)	2,753,394
N50 contigs (bp)	1,508,676
Contig L50	10
GC content contigs (%)	48.27

**Table 2 jof-09-00520-t002:** Genomic component analysis.

Attribute	Value
Total genes	11,094
Protein-coding genes	10,933
With a KOG annotation	5564
With a GO annotation	5696
With a KEGG annotation	2224
With a Swiss-Prot annotation	7027
With a Pfam annotation	7598
With a NR annotation	10,205
With a CAZy number	1133
With a PHI number	2356
With a DFVF number	2167
Secondary metabolite biosynthetic gene clusters	37
Genes encoded with secreted proteins	827
Genes with signal peptides	1057
Genes with transmembrane helices	2181
Total size (bp)	17,974,472
Average gene length (bp)	1644
Genome coding (%)	47%
Total number of ncRNA	161
Number of rRNAs	45
Number of tRNAs	73

## Data Availability

All data generated or analyzed during this study are included in this published article, its Appendix A, or deposited at the NCBI Sequence Read Archive (BioProject ID: PRJNA836725).

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
