# Peer review of "Genomic and Transcriptomic Survey Provides Insights into Molecular Basis of Pathogenicity of the Sunflower Pathogen Phoma macdonaldii"

_jof, 2023, doi:10.3390/jof9050520_

Round 1
Reviewer 1 Report
I would like to thank the authors for their efforts to do this study because there has been little attention paid to the molecular characteristics of the causal agent, P. macdonaldii (sunflower black stem), and the genomic information and genes involved in the pathogenesis of this pathogen are still unclear. In this study, the genome sequence of P. macdonaldii CXJ0811 was completed using whole-genome shotgun strategy and PacBio single molecule real-time DNA sequencing technology with high quality and completeness. Further transcription profiling analysis of various genes, including DFVF genes, PHI genes, CAZymes, transporters, secretory proteins, and a putative host-specific toxin were investigated. This study represents a comprehensive understanding of the attributes required for the plant pathogenic fungus to invade and colonize sunflower.
Comments:
Gene annotation was performed using KOG, GO, KEGG, Swiss-Prot and NR. However, each gene annotation was reported without linkage between different annotation. A total gene annotation data need to be concluded and summarized.
In phylogenetic analysis, bootstrap value cutoff should be specific and indicated on Fig 1.
Pathogenicity test was performed on independent leaves and fragments of stems. As we know the pathogenicity mechanism of the performed experiment will be different as compared with pathogenicity mechanism using plant host and pathogenic fungus. Authors concluded many results without linkage to the real pathogenic mechanism.
However, this research will offer valuable assets for genomic investigations and serve as a point of reference for controlling various species of Phoma and other related fungi.
Author Response
1.Gene annotation was performed using KOG, GO, KEGG, Swiss-Prot and NR. However, each gene annotation was reported without linkage between different annotation. A total gene annotation data need to be concluded and summarized.
Response: Thanks for the comment. According to your comments, we made modifications to the description of the data and provided a summary statement in the manuscript. We have supplemented the discussion on this part in the new submit in lines 297-305 and lines 331-334. (The gene annotation information indicated that there were 5564, 5696, 2224, 7027, and 10205 annotated genes in the KOG, GO, KEGG, Swiss-Prot and NCBI nr databases. The total number of annotated genes was 10933, and the total number of un-annotated genes was 161.)
2.In phylogenetic analysis, bootstrap value cutoff should be specific and indicated on Fig 1.
Response: Thanks for your valuable advices. We have added a bootstrap value cutoff to Figure 1, as per your suggestion and figure note has been revised.
3.Pathogenicity test was performed on independent leaves and fragments of stems. As we know the pathogenicity mechanism of the performed experiment will be different as compared with pathogenicity mechanism using plant host and pathogenic fungus. Authors concluded many results without linkage to the real pathogenic mechanism.
Response: Thanks for the comment. As a necrotrophic pathogen P. macdonaldii live and feed on dead tissue. The basic concept of necrotrophy is defined as the mode of infection in which the pathogen kills the tissue before colonization. So, we thought the DEGs profiles would probably not be affected much by independent tissues.

Reviewer 2 Report
Comments: Sunflower has high economic and ecological value. In this paper, authors sequenced the genome and Transcriptome of Phoma macdonaldii, one of the sunflower pathogens, and revealed the differences in the expression of strains in infected plants. This report provided high-quality DNA and RNA data, which provided strong support for the subsequent analysis of Phoma macdonaldii 's infection mechanism. I agree with the publication of this article after the authors carefully modified the errors involved in analyzing and writing, and especially the relevant sequencing data for publication.
Line 63, ‘Phoma sp’ should be ‘Phoma spp.’ since they are several unidentified Phoma species.
Line 105-112, Keep the font of parentheses consistent.
Line 135, Please add the version information for tRNAscan-SE and other related software.
Figure 1, How long the aligned sequence after deletion of nonhomologous and missing blocks. The confidence values of branch nodes were slightly low using NJ method.
Line 228, keep the same style for ‘N50’.
Line 231, please upload annotation file and protein file under the accession number or just attach them in SI. This is very important for other researchers.
Line 229, ‘predicated’ typo.
Table 2, Keep the same format for the numbers. Comma.
Line 243, “NCBI nr” should be better than “NR”.
Line 245, Please remove the commas in numbers in main text. It’s quite hard to read.
Line 257-258, How could you find 8719, 4597 and 2076 genes from 5696 annotated genes with GO database? Check this carefully.
Line 268, An extra parenthesis.
Line 323, There need spaces between words and marks. ‘var. saccharum’ needn’t to be italic.
Line 340, ‘one genes’ .
Line 352, Strange font of ‘p’ in ‘pathogen-host’. There are similar error in manuscript. Please check them.
Line 357-359, ‘sp.’ Don’t be italic.
Line 385, Please explain why 2d-infected stems was not included in RNA-Seq analyses.
Line 401, What does ‘|’ mean?
Figure 7b, volcano map may be a better illustration method since you take 2-days mycelium (CT) as a reference.
Line 434, There may need some space between the number and pantlessness.
Author Response
Line 63, ‘Phoma sp’ should be ‘Phoma spp.’ since they are several unidentified Phoma species.
Response: Thanks for your careful reading. The mistake has been corrected in the new manuscript in line 68.
Line 105-112, Keep the font of parentheses consistent.
Response: Thanks for your careful reading. We checked the formatting of parentheses throughout the entire manuscript. The mistake has been corrected in the new manuscript in lines 113-140.
Line 135, Please add the version information for tRNAscan-SE and other related software.
Response: Thanks indeed for this careful comment. According to your comments, We have reviewed the software used throughout the manuscript and added missing version information. We have corrected this mistake in the new manuscript lines 136-143. Furthermore, we have identified an error in which the protein families (Pfam) database was incorrectly referred to instead of the RNA families (Rfam) database at this location.
Figure 1, How long the aligned sequence after deletion of nonhomologous and missing blocks. The confidence values of branch nodes were slightly low using NJ method.
Response: Thanks for your careful reading. Our aim is to compare the phylogenetic relationships among common plant pathogenic fungi. Therefore, as the selected fungi may be distantly related, the confidence values for branch nodes were relatively low.
Line 228, keep the same style for ‘N50’.
Response: Thanks for your careful reading. The mistake has been corrected in the new manuscript in line 244.
Line 231, please upload annotation file and protein file under the accession number or just attach them in SI. This is very important for other researchers.
Response: Thanks for the comment, we have added the predicted protein and annotation files to the supplemental materials and made corresponding revisions to the order of supplemental materials mentioned in the manuscript.
Line 229, ‘predicated’ typo.
Response: Thanks for the comment, it has been revised in the new manuscript in line 242.
Table 2, Keep the same format for the numbers. Comma.
Response: According to your comments, we revised the Table 2.
Line 243, “NCBI nr” should be better than “NR”.
Response: Thanks for your careful reading. According to your comments, We revised "NR" to "NCBI nr" in the new manuscript in lines line 262.
Line 245, Please remove the commas in numbers in main text. It’s quite hard to read.
Response: According to your comments, for the sake of readability, commas were removed from numbers in the main text.
Line 257-258, How could you find 8719, 4597 and 2076 genes from 5696 annotated genes with GO database? Check this carefully.
Response: Thanks for your careful reading, we rechecked and found that data is indeed incorrectly marked. The mistake has been corrected in the new manuscript in lines 273-275. Otherwise, due to multiple GO terms being associated with a single gene, the numbers may not match up correspondingly. Additionally, each gene may also be annotated to multiple GO categories. Therefore, we have provided a revised description of the data, where the number of genes is indicated outside the parentheses, and the number of associated GO terms is indicated inside the parentheses.
Line 268, An extra parenthesis.
Response: Thanks for your careful reading, We removed the unnecessary parentheses. The mistake has been corrected in the new manuscript in line 300.
Line 323, There need spaces between words and marks. ‘var. saccharum’ needn’t to be italic.
Response: Thanks for your careful reading, The mistake has been corrected in the new manuscript in line 371.
Line 340, ‘one genes’ .
Response: Thanks for your careful reading, The mistake has been corrected in the new manuscript in line 388.
Line 352, Strange font of ‘p’ in ‘pathogen-host’. There are similar error in manuscript. Please check them.
Response: Thanks for your careful reading, The mistake has been corrected in the new manuscript in lines 400, 432 and 435.
Line 357-359, ‘sp.’ Don’t be italic.
Response: Thanks for your careful reading, The mistake has been corrected in the new manuscript in lines 305-407.
Line 385, Please explain why 2d-infected stems was not included in RNA-Seq analyses.
Response: Thanks for the comment, In the stem samples collected after two days of infection, we observed a lower fungal biomass. Considering this, we hypothesize that this might have resulted in insufficient sequencing coverage. we have supplemented the relevant elaboration in the new manuscript in lines 441-442. ( However, the lower fungal abundance observed in the stem samples inoculated after 2 days of culture may lead to insufficient sequencing coverage in the samples.)
Line 401, What does ‘|’ mean?
Response: Thanks for the comment, |Log2 (Ratio value)| ≥ 1 means absolute value of log 2 Ratio ≥ 1, For the sake of clarity, we have modified the relevant description in the article to "absolute value of log 2 Ratio ≥ 1".
Figure 7b, volcano map may be a better illustration method since you take 2-days mycelium (CT) as a reference.
Response: Thanks for your valuable advices. We followed your suggestion and modified Figure 7b to a volcano plot, along with corresponding changes in the figure caption.( b. Volcano maps display the significant DEGs between CT and each treatment group (LEAF-2d, LEAF-6d, and STEM). Blue dots represent genes that were significantly down-regulated; red dots represent genes that were significantly up-regulated; grey dots represent genes for which no significant differences were observed.and STEM.).
Line 434, There may need some space between the number and pantlessness.
Response: Thanks for your careful reading, The mistake has been corrected in the new manuscript in lines 534-538.
